# Endometrial Aging and Reproductive Decline: The Central Role of Mitochondrial Dysfunction

**DOI:** 10.3390/ijms26115060

**Published:** 2025-05-24

**Authors:** Hiroshi Kobayashi, Miki Nishio, Mai Umetani, Hiroshi Shigetomi, Shogo Imanaka, Hiratsugu Hashimoto

**Affiliations:** 1Department of Gynecology and Reproductive Medicine, Ms. Clinic MayOne, 871-1 Shijo-cho, Kashihara 634-0813, Japan; mikin.yaruki10000.boshi@gmail.com (M.N.); mai_umetani@yahoo.co.jp (M.U.); shogo_0723@naramed-u.ac.jp (S.I.); hiratsugu_hashimoto@yahoo.co.jp (H.H.); 2Department of Obstetrics and Gynecology, Nara Medical University, 840 Shijo-cho, Kashihara 634-8522, Japan; hshige35@gmail.com; 3Department of Gynecology and Reproductive Medicine, Aska Ladies Clinic, 3-3-17 Kitatomigaoka-cho, Nara 634-0001, Japan

**Keywords:** aging, endometrium, hormonal abnormalities, mitochondrial dysfunction, mitohormesis

## Abstract

Socioeconomic factors have led an increasing number of women to postpone childbirth, thereby elevating the risks of reduced fertility, pregnancy complications, preterm birth, cesarean delivery, and chromosomal abnormalities. While diminished oocyte quality is a well-established contributor to age-related infertility, endometrial dysfunction also plays a pivotal role. Optimizing both oocyte quality and endometrial health is essential for enhancing reproductive outcomes. Although aging has been defined by twelve hallmarks, research specifically addressing age-related changes in endometrial function remains limited. This review examines the process of endometrial aging, with a particular emphasis on mitochondrial function. A comprehensive literature search was conducted using PubMed and Google Scholar to identify relevant studies published up to 31 January 2025. Endometrial aging is driven by multiple biological mechanisms, most notably the decline in endometrial receptivity. Key contributing factors include hormonal dysregulation, chronic inflammation, cell cycle arrest, genomic instability, epigenetic alterations, telomere attrition, and mitochondrial dysfunction. Among these, mitochondrial dysfunction emerges as a central driver of the aging process. Endometrial senescence, precipitated by irreversible mitochondrial impairment, may underlie the progressive decline in reproductive potential. Elucidating the role of mitochondrial dysfunction in aging provides critical insights into the molecular basis of fertility decline, particularly through its impact on endometrial receptivity.

## 1. Introduction

Modern women are increasingly delaying childbearing due to social, economic, and personal factors [1]. However, age-related fertility decline has led to a growing number of women seeking infertility treatment. While assisted reproductive technologies (ARTs) can partially compensate for reduced oocyte quality, pregnancy rates remain markedly low in women over the age of 40 [2]. Ovarian aging—characterized by elevated follicle-stimulating hormone (FSH) levels and diminished oocyte quality and quantity—is a major contributor to infertility [3]. In vitro fertilization (IVF) using donor oocytes enables older women to achieve pregnancy [4,5]; however, implantation rates remain suboptimal [6], and oocyte donation is associated with the increased risks of miscarriage and pregnancy complications [7,8]. Age-related chromosomal abnormalities contribute to aneuploidy, further impairing reproductive outcomes. While preimplantation genetic testing (PGT), particularly PGT-A for aneuploidy screening, has enhanced the success of ARTs [9], older women still exhibit lower pregnancy rates [10], indicating the presence of additional age-related factors that affect implantation [11]. Advanced maternal age also elevates the risk of obstetric complications, including preeclampsia, gestational diabetes, and preterm birth [12,13]. Animal models have provided further evidence that both ovarian and endometrial aging contribute to the age-associated decline in fertility [14,15,16].

López-Otín et al. (2023) [17] identified twelve hallmarks of aging, including genomic instability, mitochondrial dysfunction, and chronic inflammation, which collectively regulate the aging process. Similarly, Wu et al. [18] defined nine core hallmarks of ovarian aging, while Tinelli et al. [19] underscored the role of uterine aging in reproductive decline, describing manifestations such as uterine atrophy, reduced blood flow, endometrial thinning, alterations in cellular composition, structural degradation of the endometrium, and aberrant molecular expression profiles. The processes of implantation, fetal development, and pregnancy maintenance are metabolically demanding, relying heavily on adequate mitochondrial function. Mounting evidence implicates mitochondrial dysfunction in age-related infertility [20,21,22], endometriosis [23,24], and polycystic ovary syndrome (PCOS) [25,26]. It has also been associated with adverse pregnancy outcomes, including fetal growth restriction (FGR) [27], preeclampsia [27,28], low birth weight [29], and preterm birth [30]. A growing body of research suggests that mitochondrial dysfunction plays a pivotal role in aging [31,32,33] and constitutes a critical mechanism underlying reproductive senescence [34,35,36]. Both mitochondrial function and endometrial receptivity decline with advancing age. Emerging evidence suggests that specific components of endometrial receptivity may be particularly vulnerable to age-related deterioration [37]. Although mitochondrial dysfunction is widely recognized as a determinant of compromised oocyte quality and suboptimal reproductive outcomes [34], its role in modulating endometrial receptivity remains inadequately elucidated.

In this review, we propose that mitochondrial dysfunction constitutes a central mechanism driving uterine aging. We examine its role in the age-related decline in endometrial function and explore its broader implications in the context of reproductive aging.

## 2. Age-Related Changes in Mitochondrial Function

This section provides a comprehensive overview of age-associated alterations in mitochondrial function and quality control mechanisms in response to cellular stress.

### 2.1. Mitochondrial Structure and Function

Human mitochondrial DNA (mtDNA) is a 16,569-base-pair circular genome encoding 37 genes essential for producing adenosine triphosphate (ATP) via oxidative phosphorylation (OXPHOS) [38]. Mitochondria metabolize glucose and fatty acids to generate ATP through the electron transport chain (ETC) [39]. In addition to energy production, mitochondria regulate key cellular processes, including metabolism, calcium and iron homeostasis, heme biosynthesis, intracellular signaling, and apoptosis [40]. ATP synthesis inherently generates reactive oxygen species (ROS), such as superoxide (O_2_^−^•), hydrogen peroxide (H_2_O_2_), and hydroxyl radicals (•OH), which, if not effectively neutralized, can induce oxidative stress and cellular damage [41]. Age-related mitochondrial decline is driven by cumulative oxidative stress, increased ROS levels, and the accumulation of mtDNA mutations, ultimately impairing ATP production and disrupting cellular homeostasis [31]. Aging is associated with functional impairments in ETC complexes I, IV, and V, as well as a heightened burden of mtDNA mutations in various tissues [42,43]. However, emerging evidence indicates that replication errors, rather than oxidative damage per se, are the primary source of age-associated mtDNA mutations [44]. Elevated heteroplasmy is pathogenic [45], and a reduced mtDNA copy number correlates with accelerated reproductive aging [46], whereas oxidative stress-induced mtDNA mutations appear to play a minimal role in this process [47]. Nonetheless, empirical data directly linking such mitochondrial alterations to aging in the human endometrium remain limited.

Cellular aging increases vulnerability to functional decline and disease [48]. In the 1950s, Harman’s free radical theory posited that ROS drive aging by inducing macromolecular damage, and this hypothesis was later refined to implicate mitochondrial ROS in the accumulation of mtDNA mutations [49,50]. Subsequent studies confirmed that mitochondrial dysfunction, oxidative stress, and altered mitochondrial dynamics are central contributors to aging and cellular senescence [20,33,51]. Notably, ROS are not exclusively deleterious; in model organisms, modest ROS elevations can activate adaptive stress responses that enhance longevity—a phenomenon termed “mitohormesis” [52,53,54]. Low ROS levels stimulate antioxidant defenses, enhance DNA repair mechanisms, and reinforce protein quality control, thereby promoting cellular resilience [54]. For example, reduced superoxide dismutase 2 (Sod2) activity in mice increases oxidative DNA damage but does not necessarily accelerate aging, suggesting that ROS may exert context-dependent protective effects [55].

### 2.2. Mitochondrial Quality Control

The maintenance of mitochondrial function requires the integration of multiple regulatory mechanisms, including nuclear–mitochondrial communication via retrograde signaling, mitochondrial dynamics (fission and fusion), autophagy, mitophagy, the mitochondrial unfolded protein response (UPR^mt^), calcium signaling, and metabolic modulation [33]. Bidirectional communication between mitochondria and the nucleus ensures cellular homeostasis, particularly under metabolic stress. In response to dysfunction or energy deficits, mitochondria initiate retrograde signaling to modulate nuclear gene expression and restore cellular function [56]. These signals include changes in mitochondrial membrane potential, elevated ROS, the altered levels of metabolites, such as ATP, nicotinamide adenine dinucleotide (NAD^+^), and acetyl coenzyme A (acetyl-CoA), and the activation of the UPR^mt^ [57,58]. Mitochondrial dynamics are regulated through coordinated fission and fusion events. Fusion is mediated by mitofusin 1 (MFN1), mitofusin 2 (MFN2), and optic atrophy protein 1 (OPA1), while fission relies on dynamin-related protein 1 (DRP1) [59]. The disruption of DRP1 impairs mitochondrial function and accelerates cellular senescence, with p53 shown to inhibit DRP1 translocation, thereby promoting aging phenotypes [60]. DRP1 deficiency leads to the elongation of dysfunctional mitochondria, whereas excessive DRP1 activity under hypoxic conditions promotes fission-mediated senescence in vascular endothelial cells [61] and cardiomyocytes [62]. An imbalance between fission and fusion underlies mitochondrial dysfunction and contributes to age-related pathologies [63]. Beyond morphological regulation, mitochondrial quality also depends on biogenesis and the selective elimination of damaged organelles [33].

### 2.3. Autophagy and Mitophagy

Autophagy is a fundamental cellular process that maintains homeostasis and mitigates stress by degrading and recycling intracellular components, including proteins, lipids, and organelles [64,65]. Mitophagy, a selective form of autophagy, is critical for mitochondrial quality control, ensuring the removal of dysfunctional or superfluous mitochondria [65,66,67]. While autophagy targets a broad range of substrates, mitophagy specifically maintains mitochondrial integrity and energy production capacity. Both autophagy and mitophagy are implicated in regulating endometrial physiology, including cyclic remodeling, embryo implantation, and decidualization [68]. Simultaneously, mitochondrial biogenesis replenishes mitochondrial pools, sustaining cellular energetics and structural integrity [62,63]. Efficient mitophagy is essential for preserving mitochondrial function, preventing ROS accumulation, and maintaining cellular viability [69]. This section examines the coordinated regulation of mitochondrial quality control pathways and their relevance to endometrial aging.

#### 2.3.1. Autophagy

Endometrial autophagy, modulated by sex hormones, is indispensable for cyclic regeneration and successful embryo implantation [68,70]. Progesterone activates autophagy via its receptor in the post-ovulatory phase [71], while the inhibition of mechanistic target of rapamycin (mTOR), particularly with rapamycin, enhances autophagy, suppresses cellular senescence, and extends lifespan in various model organisms, including yeast, nematodes, Drosophila, and mice [72,73,74]. Autophagy is regulated by nutrient-sensing pathways involving insulin/insulin-like growth factor-1 (IGF-1), AMP-activated protein kinase (AMPK), sirtuin 1 (SIRT1), and the phosphatidylinositol-3-kinase (PI3K)/AKT/mTOR axis. Deficiencies in AMPK and SIRT1 have been associated with endometrial fibrosis, infertility, and reduced uterine functionality [75,76,77]. The dysregulation of PI3K/AKT/mTOR signaling in the aging endometrium may compromise implantation success [78]. Autophagy is essential for decidualization and implantation; its impairment has been linked to infertility [79,80]. Pharmacologic and genetic interventions using rapamycin, chloroquine, and autophagy inhibitors underscore the critical role of autophagy in reproductive biology [81,82,83]. Defective autophagy is also associated with implantation failure, pregnancy complications, and placental abnormalities [77,80,84,85,86]. Enhancing autophagy may offer therapeutic potential to prevent age-associated reproductive decline [80,87].

#### 2.3.2. Mitophagy

Mitophagy, particularly via the PTEN-induced kinase 1 (PINK1)/Parkin pathway, selectively eliminates damaged mitochondria to preserve cellular integrity [88,89]. Upon mitochondrial dysfunction, PINK1 accumulates on the outer mitochondrial membrane, recruiting Parkin to ubiquitinate defective mitochondria, targeting them for lysosomal degradation. The disruption of the fission–fusion balance impairs ATP production, elevates ROS levels, and accelerates cellular aging [63]. Mitophagy supports both oocyte [90] and endometrial health [91] and plays a role in endometrial regeneration and implantation; its impairment contributes to infertility [70,92,93]. Aberrant mitochondrial dynamics increase miscarriage risk, and defective mitophagy exacerbates oxidative stress and energy deficits, heightening susceptibility to endometriosis and infertility [94,95,96]. Mitophagy-enhancing agents have been shown to prolong lifespan and delay age-associated diseases in animal models [97,98]. Maintaining a balance between mitophagy and mitochondrial biogenesis is critical for cellular function, and targeting mitophagy represents a promising strategy for improving fertility and mitigating reproductive aging.

## 3. Age-Related Regulation of Endometrial Receptivity

The endometrium comprises a heterogeneous population of cells, including luminal and glandular epithelial cells, stromal cells, stem cells, immune cells, and endothelial cells. Its cyclical and dynamic regenerative capacity throughout the menstrual cycle is intricately linked to reproductive competence. A minimum endometrial thickness is essential for successful implantation, and women with a thin endometrium (≤7 mm) are generally older than those with a thicker endometrium (>7 mm) [99]. Ultrasonographic studies indicate that 61% of women over the age of 40 exhibit an endometrial thickness of less than 10 mm, compared to only 29% of younger women, with an average reduction of approximately 0.52 mm in endometrial thickness in women aged 35–40 and above [100]. The age-related thinning of the endometrium is associated with reduced implantation and pregnancy rates, as well as the increased rates of miscarriage [19,100]. Vascular stiffening due to atherosclerosis impairs uterine blood flow, negatively impacting endometrial growth and receptivity [101]. In parallel, the age-associated decline in estrogen levels further contributes to endometrial thinning and reduced perfusion [100]. Moreover, the aging endometrial luminal epithelium shows a higher proportion of ciliated epithelial cells. While these cells facilitate sperm transport, they may also disrupt the local balance of secretory factors necessary for embryo attachment, thereby impairing implantation potential [102,103,104].

The processes of repair, proliferation, secretion, and differentiation in the principal endometrial cell types—luminal epithelial cells, glandular epithelial cells, and stromal cells—are stringently regulated by estrogen and progesterone. These hormonally driven events are energetically sustained by mitochondrial ATP synthesis [105]. Mitochondrial estrogen receptor beta (ERβ) plays a pivotal role in this context by modulating calcium influx, ATP synthesis, apoptotic signaling, and reactive oxygen species (ROS) generation, thereby contributing to the maintenance of endometrial receptivity [106]. The establishment of receptivity involves angiogenesis, stromal edema, and secretory transformation, while successful implantation requires the coordinated regulation of adhesion, proliferation, metabolism, and immune tolerance [107]. Hormonal stimulation influences both mitochondrial and nuclear gene expression associated with endometrial receptivity, including ERβ [108,109], homeobox A10 (HOXA10) [110], leukemia inhibitory factor (LIF) [111], integrin αVβ3 [112], glycodelin [113], and connexin 43 (Cx43) [114]. Age-related mitochondrial dysfunction downregulates the expression of these genes, leading to diminished estrogen responsiveness, impaired endometrial receptivity, and decreased fertility [92,93,115]. Collectively, these findings highlight the critical role of mitochondrial function in mediating the hormone-dependent regulation of endometrial receptivity. Interventions aimed at preserving mitochondrial health and enhancing cellular energy metabolism may represent promising therapeutic strategies to improve endometrial function and fertility outcomes in aging women. Senescent alterations in the endometrium influence all cellular constituents, including stromal, glandular, and luminal epithelial cells, potentially compromising the tissue’s regenerative potential and fecundity. Among these, stromal cells are pivotal for maintaining structural integrity and physiological function and are particularly vulnerable to age-associated deterioration. In this study, the term “endometrium” refers to both stromal and epithelial components unless specified otherwise.

## 4. Characteristics of Endometrial Aging

Aging affects both the uterus and ovaries, influencing pregnancy outcomes even in assisted reproductive technologies such as egg donation and in vitro fertilization (IVF). Uterine aging contributes significantly to fertility decline, with advanced maternal age associated with higher miscarriage rates, even in donor oocyte cycles [116,117,118]. Endometrial aging is characterized by cellular senescence, defined by irreversible cell cycle arrest and a proinflammatory secretory phenotype driven by DNA damage, oxidative stress, and telomere shortening [119,120]. In light of the 2023 updated hallmarks of aging [17], we propose the inclusion of hormonal abnormalities as a key hallmark of endometrial aging. This section examines the role of mitochondrial dysfunction in endometrial aging and its implications for reproductive health.

### 4.1. Hormonal Abnormalities

Endometrial physiology is tightly regulated by hormones and signaling molecules originating from the hypothalamus, pituitary gland, ovaries, adrenal glands, adipose tissue, and thyroid. Gonadotropin-releasing hormone (GnRH) from the hypothalamus stimulates the release of follicle-stimulating hormone (FSH) and luteinizing hormone (LH) from the anterior pituitary, which in turn regulate ovarian estrogen and progesterone production—both essential for endometrial proliferation and decidualization [121]. Additionally, the endometrium itself secretes important implantation-related factors such as prolactin and insulin-like growth factor-binding protein 1 (IGFBP1) [121]. Aging disrupts this hormonal network, leading to reduced levels of estrogen and progesterone, impaired endometrial function, and diminished fertility. Estrogen receptors (ERs) are localized in the nucleus, cytoplasm, and mitochondria of endometrial cells, emphasizing the hormone’s multifaceted regulatory roles. Age-related endocrine changes adversely affect endometrial physiology, as evidenced by experimental rodent models showing a progressive decline in ER expression with age [122]. Elevated FSH levels have also been associated with endometrial atrophy [123].

Mitochondria are central to steroid hormone biosynthesis and metabolism, particularly by regulating cholesterol transport—the rate-limiting step in steroidogenesis—through the steroidogenic acute regulatory protein (StAR) [124] (Figure 1). Within mitochondria, cholesterol is converted to pregnenolone by cytochrome P450 side-chain cleavage enzyme (CYP11A1), the precursor for all steroid hormones [125]. Mitochondrial dynamics, particularly the balance between fusion and fission, are essential for effective steroidogenesis, as mitochondrial fusion enhances electron transport chain (ETC) efficiency, ATP production, and P450 enzyme activity, all critical for hormone synthesis [126]. Estrogen supports mitochondrial integrity by modulating gene expression and oxidative phosphorylation (OXPHOS) through estrogen receptor subtypes ERα and ERβ [127]. Moreover, it promotes mitochondrial function by upregulating fusion-related proteins (MFN1, MFN2, OPA1) and regulating fission-associated proteins such as dynamin-related protein 1 (DRP1) and fission, mitochondrial 1 (FIS1) [128]. Estrogen also plays a key role in promoting mitophagy—the selective autophagic removal of damaged mitochondria. Estrogen deficiency impairs mitophagy, leading to the accumulation of dysfunctional mitochondria and increased cellular stress [94]. Additionally, estrogen-related receptor alpha (ERRα) interacts with peroxisome proliferator-activated receptor gamma coactivator-1 alpha (PGC-1α) to maintain mitochondrial function and mitigate oxidative damage [128]. By upregulating superoxide dismutase 2 (SOD2), estrogen enhances antioxidant defenses and preserves mitochondrial integrity [128]. Through these mechanisms, estrogen supports cellular bioenergetics, promotes mitochondrial biogenesis, maintains metabolic homeostasis, and strengthens antioxidant capacity. Consequently, the decline in estrogen during the menopausal transition leads to mitochondrial dysfunction, elevated oxidative stress, and impaired mitochondrial dynamics [129].

Mitochondrial dysfunction and hormonal dysregulation are deeply interconnected in the aging process. Aged endometrial stromal cells exhibit reduced sensitivity to progesterone, which may contribute to age-related infertility [92]. Given the indispensable role of mitochondria in estrogen biosynthesis and the reciprocal protective effects of estrogen on mitochondrial function, a decline in mitochondrial efficiency fosters hormonal imbalances. This creates a deleterious feedback loop that accelerates endometrial aging. The bidirectional relationship between steroid hormones and mitochondrial function underscores their central role in the aging of the endometrium and the broader context of reproductive aging. Thus, estrogen deficiency disrupts mitochondrial function through mechanisms including oxidative stress, impaired mitochondrial dynamics, and decreased mitophagy, ultimately reducing endometrial receptivity and contributing to reproductive senescence.

### 4.2. Chronic Inflammation

Chronic inflammation is closely associated with cellular senescence, a phenomenon often described as inflammatory senescence, wherein the elevated levels of proinflammatory cytokines—such as interleukin-1β (IL-1β)—induce senescence in endometrial stromal cells [130]. This inflammatory microenvironment is largely shaped by the senescence-associated secretory phenotype (SASP), a collective term used for cytokines, chemokines, growth factors, proteases, and extracellular vesicles (EVs) [131] (Figure 2). Key SASP components include proinflammatory cytokines (e.g., interleukin-6 [IL-6], IL-1β, IL-8), chemokines (e.g., C-X-C motif chemokine ligand 8 [CXCL-8/IL-8]), growth factors (e.g., transforming growth factor-beta [TGF-β], insulin-like growth factor 1 [IGF-1], vascular endothelial growth factor [VEGF]), matrix metalloproteinases (MMPs), and exosomes. The SASP is tightly regulated by hypoxia-inducible factor 1-alpha (HIF-1α) and the nuclear factor-kappa B (NF-κB) signaling pathway, both critical in aging-associated inflammation. HIF-1α, a central mediator of cellular responses to hypoxia, regulates IL-1β expression and is implicated in the pathogenesis of atherosclerosis [132]. Under normoxic conditions, HIF-1α is degraded, but under hypoxia, it stabilizes and modulates cell proliferation and differentiation. It also interacts with NF-κB and signal transducer and activator of transcription 3 (STAT3), both of which are involved in aging-related pathophysiology [132]. NF-κB is activated in response to oxidative stress and DNA damage, promoting SASP factor transcription [133]. The DNA damage response (DDR) further amplifies SASP activation by stabilizing GATA-binding protein 4 (GATA4), a transcription factor that enhances NF-κB activity [134]. Autophagy, mediated by sequestosome 1 (SQSTM1/p62), inhibits GATA4 degradation, thereby sustaining NF-κB signaling. The mTOR pathway also regulates SASP by promoting IL-1α translation and maintaining SASP activity through a feedback loop with NF-κB [135]. Epigenetic changes, such as the demethylation of histone H3 lysine 27 trimethylation (H3K27me3), further enhance SASP gene expression [136].

SASP factors propagate senescence in adjacent cells, reduce endometrial plasticity, and perpetuate chronic inflammation—an environment strongly associated with mitochondrial dysfunction [131]. Conversely, mitochondrial dysfunction intensifies inflammation by increasing reactive oxygen species (ROS) and promoting the release of mitochondrial DNA (mtDNA) into the cytoplasm or extracellular space [33]. Extracellular mtDNA acts as a damage-associated molecular pattern (DAMP), activating inflammasomes and cytosolic DNA sensors, thereby initiating inflammatory cascades [33]. This interplay among chronic inflammation, SASP, and mitochondrial dysfunction establishes a self-perpetuating, proinflammatory cycle [137]. While transient SASP signaling may contribute to tissue repair, persistent SASP disrupts regeneration and impairs stem cell function. Interestingly, mild inflammation induced by endometrial biopsy has been associated with improved implantation outcomes, whereas excessive inflammation compromises endometrial receptivity [138].

Chronic inflammation is intrinsically linked to aging. Inflammation-related pathways, such as IL1A and interferon (IFN) signaling, are upregulated in endometrial cells derived from aged bovine models—a mixed epithelial and stromal cell culture—compared to younger counterparts [139]. In humans, chronic inflammation contributes to age-related fertility decline by disrupting uterine immune homeostasis [140]. SASP-driven intercellular signaling promotes stem cell exhaustion, reduces regenerative capacity, and accelerates cellular senescence. The accumulation of senescent cells exacerbates inflammation—a process termed inflammaging—which accelerates aging [140]. Mitochondrial dysfunction, a hallmark of aging, contributes to this process by inducing mtDNA mutations, destabilizing respiratory complexes, and triggering apoptosis [33,141]. The cytosolic leakage of mtDNA activates the cyclic GMP–AMP synthase–stimulator of interferon genes (cGAS–STING) pathway, stimulating type I interferon and cytokine secretion. This pathway plays a critical role in age-related inflammation and neurodegeneration, largely through the activation of the NLRP3 inflammasome [142]. Furthermore, the cGAS–STING pathway has been implicated in amyloid-beta accumulation in Alzheimer’s disease and reduced uterine receptivity in aged murine models [143]. Its involvement in reproductive aging suggests a possible contribution to infertility in older women [144].

The small GTPase CDC42, implicated in endometrial senescence, is also linked to the SASP and inflammatory responses [145]. The downregulation of CDC42 is associated with defective decidualization and impaired uterine receptivity in patients with recurrent implantation failure, likely via the aberrant activation of the Wnt signaling pathway [146]. Moreover, in primary human endometrial cell cultures comprising both epithelial and stromal populations, senescence induction results in the downregulation of *SIRT1*, *SIRT6*, and telomeric repeat-binding factor 1 (*TERF1*), accompanied by the elevated levels of IL-6, IL-8, IL-1α, matrix metalloproteinase 3 (MMP3), and p16 [131]. *TERF1* is essential for maintaining telomere integrity. These findings underscore the close association between endometrial senescence and SASP expression [131]. In reproductive tissues such as the endometrium, the accumulation of senescent cells and SASP components disrupts tissue homeostasis. The resulting chronic inflammatory environment and mitochondrial dysfunction impair endometrial receptivity, which is essential for successful implantation and pregnancy. Despite these advances, the precise molecular mechanisms underlying SASP secretion in the endometrium remain incompletely understood and represent a critical area for future investigation. Ultimately, chronic inflammation and mitochondrial dysfunction form a bidirectional axis that sustains SASP activation, inflammasome stimulation, and apoptotic signaling. This proinflammatory cascade accelerates reproductive aging and impairs fertility potential [130], highlighting chronic inflammation as a key contributor to age-related infertility [147].

### 4.3. Cell Cycle Arrest

As endometrial stromal cells undergo senescence, the expression of key senescence markers—including *cyclin-dependent kinase inhibitor 2A* (*CDKN2A*, encoding p16^INK4a^), *CDKN1A* (p21^Cip1/Waf1^), *NOTCH*, and senescence-associated β-galactosidase (SA-β-Gal)—is markedly upregulated [120] (Figure 3). p16^INK4a^ functions as a cyclin-dependent kinase (CDK) inhibitor that suppresses CDK4/6 activity, thereby inducing cell cycle arrest, promoting cellular senescence, and preventing the proliferation of damaged cells [148]. Importantly, *CDKN2A* encodes two distinct proteins—p16^INK4a^ and the tumor suppressor ARF (alternative reading frame)—via alternative splicing and translation initiation, each possessing unique structural and functional characteristics [149]. While ARF contributes to TP53 stabilization, it also plays a role in regulating autophagy, underscoring its importance in maintaining cellular homeostasis [149,150]. Another critical function of p16^INK4a^ involves the suppression of mitochondrial biogenesis and the modulation of mitochondrial membrane potential [151]. The dysregulation of these quality control mechanisms—often associated with elevated *CDKN2A* expression—can lead to the accumulation of dysfunctional mitochondria, thereby impairing cellular proliferation and hormonal responsiveness. Such alterations diminish endometrial receptivity and negatively affect implantation potential [92,100].

The Notch signaling pathway also plays a pivotal role in regulating cell cycle progression within the epithelial secretory lineage of the endometrium by facilitating the G1-to-S phase transition and influencing the G2/M transition [102,152]. Notably, Notch1 localizes to mitochondria, where it directly modulates mitochondrial function and dynamics [153]. Through metabolic reprogramming, Notch signaling significantly impacts OXPHOS and glycolysis [153]. Increased Notch activity enhances glycolytic flux while suppressing OXPHOS, thereby affecting mitochondrial efficiency and energy production [154]. Notch signaling is also essential for maintaining stemness via mitochondrial regulation, a process fundamental to cell fate determination and inflammatory responses [155]. Thus, both *CDKN2A* [151] and Notch signaling [156] are intricately linked to mitochondrial homeostasis. Furthermore, *NOTCH* genes are central to cellular development and tissue homeostasis—processes that are profoundly influenced by aging. Alongside *CDKN2A*, age-related alterations in Notch signaling have been implicated in various senescence-associated changes [148,157]. For example, in aged epidermal tissue, mutations in these gene networks drive clonal cell expansion and enhance cellular survival under stress conditions [158]. Senescent cells typically exhibit mitochondrial dysfunction, characterized by increased ROS production and a metabolic shift from OXPHOS toward glycolysis [159]. Given the cyclical regeneration of the endometrium in preparation for implantation, this tissue is particularly vulnerable to age-associated senescence. The accumulation of senescent cells—driven by the SASP and upregulation of *CDKN2A* and *NOTCH*—likely represents an adaptive response to aging. However, while this adaptation may provide temporary protection against environmental stressors, it ultimately impairs uterine function and reduces fertility potential. Advancing maternal age is associated with reduced glucose uptake in endometrial tissue and the decreased activity of mitochondrial enzymes such as succinate dehydrogenase (complex II) and cytochrome c oxidase (complex IV), both of which are essential components of the electron transport chain (ETC) within the inner mitochondrial membrane [160]. A study investigating molecular factors contributing to impaired endometrial receptivity in women of advanced maternal age (AMA) reported elevated *CDKN2A* expression and a pronounced molecular interplay between p16 and the Notch signaling pathway—a key regulator of cell cycle progression and apoptosis [102]. Under physiological conditions, Notch activation promotes apoptosis in multiciliated cells (MCCs); however, in AMA women, an aberrant increase in MCCs is observed, likely due to impaired Notch-mediated apoptosis and dysregulated estrogen/progesterone signaling [102]. The accumulation of senescent p16-positive cells suggests a shift in cell fate determination that favors the expansion of MCCs—a cell population suboptimal for embryo implantation [102]. Collectively, the overexpression of *CDKN2A/p16* and the disruption of Notch signaling contribute to imbalances in epithelial cell composition, mitochondrial dysfunction, and diminished endometrial receptivity with age. These findings underscore their central roles in the molecular pathogenesis of endometrial aging.

### 4.4. Genomic Instability

Genomic instability is characterized by the accumulation of DNA mutations, chromosomal abnormalities, and copy number variations, which progressively increase with age and ultimately lead to irreversible growth arrest [92,161,162]. In the endometrium, such instability contributes to functional decline and reduced receptivity due to the accumulation of senescent cells and the secretion of proinflammatory mediators. These changes negatively impact fertility and elevate the risk of endometrial malignancies [115,131,163]. The endometrium is particularly susceptible to oxidative stress, partly due to the presence of free iron in menstrual blood, which catalyzes the Fenton reaction and induces DNA damage [164] (Figure 4). Oxidative stress is also implicated in pathologies such as endometriosis [165] and ovarian cancer [166], highlighting the critical importance of iron regulation in reproductive health. ROS damage both nuclear and mitochondrial DNA (e.g., formation of 8-oxoguanine [8-oxoG]) and compromise the integrity of the mitochondrial permeability transition pore (mPTP), thereby accelerating disturbances in mitochondrial dynamics and apoptotic regulation [167,168]. Mitochondrial degradation subsequently reduces respiratory capacity and membrane potential, exacerbating ROS accumulation. Thus, preserving mitochondrial energy homeostasis is essential for maintaining genomic stability. In response to nuclear DNA damage, cells activate a network of DNA repair enzymes, including p53, ataxia telangiectasia mutated (ATM), and ATM- and Rad3-related (ATR) kinases [169]. However, mutations in nuclear DNA can disrupt genes regulating autophagy and mitochondrial dynamics (e.g., *PINK1*, *PARKIN*), leading to defective mitophagy and the breakdown of mitochondrial quality control mechanisms [170]. As mitochondrial stress intensifies, further ROS accumulation inflicts additional damage on DNA, repair enzymes, and regulatory genes [171]. Excessive ROS also inactivates 8-oxoguanine DNA glycosylase 1 (OGG1), a key enzyme in the base excision repair pathway, and activates proinflammatory transcription factors such as NF-κB [172]. Mutations in *BRCA1*—a gene essential for homologous recombination-mediated DNA repair—are associated with diminished ovarian reserve [173] and premature menopause [174], suggesting that impaired DNA repair accelerates reproductive aging. Additionally, sirtuin family genes such as *SIRT1* and *SIRT6* are crucial for maintaining genomic integrity. The age-related downregulation of these genes compromises both DNA repair processes and mitochondrial function [175]. Notably, *SIRT6*-deficient mice exhibit phenotypes consistent with accelerated aging, underscoring its vital role [175].

These findings indicate that mitochondrial stress, driven by ROS overproduction, contributes to nuclear DNA damage, chromosomal aberrations, cellular senescence, and carcinogenesis—highlighting the role of mitochondrial dysfunction in destabilizing the genome. Conversely, genomic instability exacerbates mitochondrial and nuclear stress, leading to mutations in both nuclear and mitochondrial DNA, impaired autophagy, and disrupted DNA repair mechanisms. This results in the accumulation of dysfunctional mitochondria, reduced ATP production, and sustained ROS generation. Consequently, genomic instability and mitochondrial dysfunction form a self-reinforcing negative feedback loop, each aggravating the other. This interplay underlies the pathogenesis and progression of numerous age-related conditions, including infertility, cancer, and neurodegenerative diseases [176,177]. In the context of reproductive health, genomic instability contributes to germ cell mutations and meiotic abnormalities through damage to both nuclear and mitochondrial DNA, resulting in infertility and diminished ovarian function [178]. The interplay between age-associated DNA damage, impaired mitochondrial function, defective DNA repair, and genomic instability is central to reproductive aging. Moreover, mitochondrial dysfunction exacerbates genomic instability by increasing oxidative stress and accelerating telomere attrition [102]. Senescent cells commonly exhibit elevated ROS levels, which damage nuclear and mitochondrial DNA. Mutations in mitochondrial DNA can impair ATP synthesis, increase the risk of aneuploidy, and ultimately reduce the efficacy of assisted reproductive technologies (ARTs) in women of advanced maternal age [20].

### 4.5. Epigenetic Modifications

Aging profoundly influences DNA methylation and histone acetylation patterns, as well as the expression of enzymes involved in acetylation, such as histone acetyltransferases (HATs) and histone deacetylases (HDACs). Histone acetylation promotes chromatin relaxation, thereby facilitating gene transcription. HATs catalyze the transfer of acetyl groups to histones using acetyl-CoA as a donor, modulating gene expression in a context-dependent manner. Age-related alterations in histone acetylation are tissue-specific: histone H3 acetylation (AcH3) increases in aged skin [179], whereas in the brain, aging is associated with the reduced acetylation of histone H3 at lysine 27 (H3K27ac) [180]. Endometrial aging exerts its effects across all major cellular compartments—luminal epithelial, glandular epithelial, and stromal cells—via distinct epigenetic modifications, including alterations in DNA methylation and histone modification patterns [181]. In the endometrium, the acetylation of histones H3K9 and H4K8 increases during the early proliferative phase, supporting endometrial regeneration, while H4K8 acetylation peaks during the early secretory phase and declines in the late secretory phase, correlating with reduced endometrial receptivity [181] (Figure 5). Sirtuins, particularly the NAD^+^-dependent deacetylase SIRT1, play key roles in DNA repair, metabolic regulation, and the enhancement of mitochondrial function and energy efficiency [182] and are closely associated with longevity [163]. Age-related changes in DNA methylation and histone modifications contribute to the decreased expression of critical genes such as *HOXA10*, which is essential for endometrial receptivity; its downregulation has been linked to implantation failure [183]. Moreover, epigenetic alterations impair the expression of genes required for decidualization [184], and the age-related disruption of these epigenetic patterns contributes to a decline in fertility [163,183,184].

In addition to nuclear DNA, the epigenetic modifications of mitochondrial DNA (mtDNA) contribute to mitochondrial energy deficiencies and increased oxidative stress, which negatively affect endometrial receptivity and overall fertility [185]. Although mtDNA was previously believed to lack epigenetic regulation due to the absence of histones, recent evidence has revealed that mechanisms such as DNA methylation and the action of non-coding RNAs modulate mtDNA expression and function [186]. Mitochondria also play a central role in epigenetic regulation by supplying key metabolites—such as S-adenosylmethionine (SAM), flavin adenine dinucleotide (FAD), α-ketoglutarate (α-KG), acetyl-CoA, methionine, folate (vitamin B9), cobalamin (vitamin B12), and lactate—that are essential for DNA methylation and histone acetylation [187,188]. The production of these metabolites occurs in various cellular compartments, with mitochondria playing a pivotal role in generating intermediates like acetyl-CoA and α-KG. In aged female mice, specific CpG islands within critical decidualization genes, such as *Hoxa10*, undergo hypermethylation, potentially impairing decidualization and reducing fertility during the implantation window [184]. These mitochondria-derived metabolites directly influence epigenetic processes, thereby linking cellular metabolic status to gene regulation. Consequently, the interplay between age-associated mitochondrial dysfunction and epigenetic dysregulation compromises endometrial receptivity, disrupts decidualization, and impairs fertility. Notably, the hypermethylation of the *HOXA10* promoter has been identified in several gynecologic conditions associated with infertility, including endometriosis, endometrial polyps, and submucosal fibroids [189]. This epigenetic modification represses *HOXA10* expression, emphasizing its pivotal role in impaired decidualization and reduced fertility in humans [189].

### 4.6. Telomere Attrition

Telomeres are specialized nucleoprotein structures that protect the terminal regions of chromosomes and progressively shorten with each cell division. When telomeres become critically short, this truncation triggers cell cycle arrest; however, telomere length is maintained in part by the enzyme telomerase. In the human endometrial epithelium, telomerase expression and activity fluctuate in response to the ovarian hormone cycle, with the lowest activity observed during the progesterone-dominant phase [190]. Variations in telomerase activity throughout the menstrual cycle are largely confined to the epithelial compartment of the endometrium, whereas stromal cells maintain persistently low activity levels irrespective of hormonal fluctuations. In senescent cells, shortened telomeres are more susceptible to DNA damage, which contributes to genomic instability (Figure 6). Telomere attrition and epigenetic modifications are intricately interconnected processes that together influence cellular aging and genomic integrity [191,192]. Senescent cells exhibit reduced levels of histones H3 and H4, leading to a relaxed chromatin structure that enhances transcriptional accessibility but also increases transcriptional noise [193]. In endometrial tissue, the dysregulation of histone acetylation—particularly involving histones H3 and H4—has been implicated in implantation failure, suggesting a role for histone modifications in endometrial remodeling throughout the menstrual cycle, as well as in age-related pathologies [194]. During senescence, levels of SIRT1 and acetylation of histone H4 at lysine 16 (H4K16ac) decline, contributing to chromatin decompaction [195]. SIRT1, a NAD^+^-dependent deacetylase, specifically targets H4K16ac and suppresses its acetylation [195]. In murine models, a decline in SIRT1 expression has been observed in the uterine decidua of aged mice, implicating SIRT1 in uterine aging and decreased reproductive capacity [77]. The expression of all sirtuin isoforms in the human endometrium [196] further supports a role for SIRT1 in facilitating embryo implantation and maintaining uterine receptivity. SIRT1 is expressed in all principal cellular constituents of the human endometrium—luminal epithelial cells, glandular epithelial cells, and stromal cells—with markedly elevated expression observed in stromal cells. Additionally, senescence is accompanied by the decreased levels of H3K27me3, a repressive histone mark, leading to the activation of genes such as *p16* and other senescence-associated secretory phenotype (SASP) genes, thereby promoting the senescent state [197]. Conversely, levels of H3K4me3, a histone mark associated with gene activation, are diminished in senescent cells [197]. These post-translational histone modifications—methylation and acetylation—affect the chromatin structure in telomeres, influencing gene expression and contributing to genomic instability and the activation of transposable elements [198]. Conditional knockout studies of *Ezh2* (a component of the polycomb repressive complex 2) further underscore the critical role of Ezh2-mediated H3K27me3 in supporting successful implantation [199].

Furthermore, aging-related oxidative stress, primarily driven by mitochondrial dysfunction, exacerbates telomere attrition [200]. The inhibition of nucleotide synthesis, essential for DNA repair, also accelerates telomere shortening [201]. Critically short telomeres activate the p53 pathway, which suppresses the expression of *PGC-1α* and *PGC-1β*, leading to impaired mitochondrial function and creating a self-perpetuating cycle of cellular aging [202]. *PGC-1α* governs mitochondrial biogenesis by coactivating transcription factors such as nuclear respiratory factors 1 and 2 (NRF1 and NRF2) and mitochondrial transcription factor A (TFAM) [203]. It also interacts with nuclear receptors such as estrogen-related receptor α (ERRα), regulates mitochondrial dynamics, and is itself modulated by SIRT1-dependent deacetylation [203]. In summary, telomere attrition inhibits mitochondrial biogenesis, amplifies oxidative stress, and accelerates reproductive aging by impairing mitochondrial recovery mechanisms [204,205,206]. In endometrial cells, these mitochondrial impairments disrupt energy-dependent processes essential for implantation and decidualization, ultimately reducing fertility. In murine models, telomere shortening leads to increased embryo fragmentation and arrest, meiotic spindle defects, and genomic instability [207]. In humans, shortened telomeres have been observed in oocytes from women experiencing IVF failure and in fragmented or aneuploid embryos [207]. However, the relationship between telomere shortening and infertility remains complex; paradoxically, elongated telomeres have been reported in certain cases of endometriosis and premature ovarian failure [206]. Stem cells, which maintain long telomeres via telomerase, have been hypothesized to contribute to the pathogenesis of endometriosis. Thus, the association between telomere dynamics and reproductive disorders is nuanced and multifactorial. Further research is required to elucidate the precise impact of telomere length on fertility.

### 4.7. MicroRNA Changes

MicroRNAs (miRNAs) are small non-coding RNAs that regulate gene expression at the post-transcriptional level. They are involved in numerous reproductive processes, including oocyte maturation and spermatogenesis [208]. miRNAs are also integral to the regulation of endometrial receptivity, exerting their effects across luminal epithelial, glandular epithelial, and stromal cell types. Increasing evidence suggests that miRNAs also play critical roles in regulating both mitochondrial function and endometrial receptivity. Specific miRNAs, such as miR-124-3p and miR-152-3p, exhibit expression patterns that fluctuate in response to progesterone levels [209]. For example, miR-1203 targets cyclophilin D (CypD), promoting the closure of the mitochondrial permeability transition pore (mPTP), thereby protecting both epithelial and stromal endometrial cells from oxidative stress [210]. The altered expression of several miRNAs—including miR-30b, miR-30d, and miR-494—has been detected in endometrial biopsies during the implantation window, suggesting their involvement in modulating endometrial receptivity [211]. These miRNAs regulate key genes required for successful embryo implantation, such as *calpastatin* (*CAST*), *cystic fibrosis transmembrane conductance regulator* (*CFTR*), *fibroblast growth factor receptor 2* (*FGFR2*), and *leukemia inhibitory factor* (*LIF*). For instance, miR-30d plays a role in embryo–endometrial adhesion, while miR-29c and miR-125b have been found to be dysregulated in infertile women [212]. miR-31 is significantly elevated in endometrial tissue during the secretory phase compared to the proliferative phase [213], suggesting a role in preparing the endometrium for potential implantation. However, contrasting findings have identified miR-31 as a negative biomarker of endometrial receptivity [208]. Furthermore, the overexpression of miR-124-3p has been associated with reduced embryo implantation rates [214,215]. Similarly, miR-27a-3p, which is upregulated in chronic endometritis, has been implicated in reduced uterine receptivity. miR-200c negatively affects receptivity by targeting *fucosyltransferase 4* (FUT4) [216]. Collectively, these findings highlight that altered miRNA expression can significantly influence endometrial receptivity in humans.

Age-related factors—including hormonal fluctuations, epigenetic changes, oxidative stress, the accumulation of DNA damage, SASP, chronic inflammation, and alterations in miRNA biogenesis—contribute to the dysregulation of miRNA expression [217]. With advancing age, the expression levels of miR-223-3p, miR-155-5p, and miR-129-5p are altered, while the dysregulation of miR-181c is closely associated with mitochondrial dysfunction [218]. A subset of miRNAs, collectively referred to as mitochondrial miRNAs (mitomiRs), localize within mitochondria and regulate mitochondrial gene expression and function [219]. Although encoded by nuclear DNA, these miRNAs are translocated into mitochondria, where they influence the expression of mitochondrial DNA (mtDNA) and mitochondria-related genes. It has been proposed that age-related alterations in endometrial miRNA expression may impair endometrial receptivity by disrupting mitochondrial function. However, direct evidence linking age-related miRNA changes to impaired endometrial receptivity in humans remains limited.

### 4.8. Loss of Protein Homeostasis

Mitochondria possess an intrinsic protein quality control system that plays a pivotal role in maintaining protein homeostasis (proteostasis) [220]. Proteostasis encompasses the regulation of protein synthesis, folding, and degradation and involves key components such as heat shock proteins (HSPs), the ubiquitin–proteasome system (UPS), and mitochondrial Lon proteases [221]. Human endometrial cells ubiquitously express a broad spectrum of heat shock proteins (HSPs), including HSP27, HSP60, HSP70, and HSP90, across luminal epithelial cells, glandular epithelial cells, and stromal cells. The expression of these proteins fluctuates throughout the menstrual cycle, indicating hormonal regulation [222]. HSPs are critically involved in reproductive processes such as decidualization, implantation, and placentation. The dysregulation of HSP expression has been linked to implantation failure, pregnancy loss, and other fetomaternal complications [223]. The UPS is essential for maintaining cellular homeostasis in human endometrial cells and plays a vital role in supporting endometrial function. It facilitates the degradation of estrogen receptors and regulates cell cycle progression and apoptosis—processes that are fundamental to fertility [224]. Furthermore, the reduced expression of LONP1 has been observed in the oocytes of aged animals, implicating it in oocyte aging [225].

Mitochondrial function and proteostasis are closely interconnected. Mechanisms such as the mitochondrial unfolded protein response (UPR^mt^), mitophagy, and ribosomal function contribute to the regulation of this balance [226]. Age-related increases in ROS lead to damage of both mitochondrial DNA (mtDNA) and proteins [96,227], promoting the accumulation of misfolded proteins [228]. Impaired mitophagy results in the accumulation of dysfunctional mitochondria, further exacerbating mitochondrial dysfunction and disrupting proteostasis [227,229]. In addition, diminished ribosomal function contributes to increased translational errors and the accumulation of misfolded proteins, further destabilizing proteostasis [48]. These alterations may contribute to the decline in mitochondrial function and accelerate cellular senescence [230], potentially reducing endometrial receptivity and increasing the risk of infertility [19]. The endometrium undergoes cyclical morphological and molecular changes to prepare for embryo implantation. Disruptions in protein expression and secretion can impair endometrial receptivity, leading to infertility. Altered proteostasis may also influence oxidative stress responses and hormonal signaling, thereby contributing to diminished reproductive function. However, the direct impact of proteostasis dysregulation on the human endometrium remains to be fully elucidated.

### 4.9. Stem Cell Depletion

The periodic shedding and regeneration of the endometrium are processes fundamentally dependent on endometrial stem cells [19]. The self-renewal and differentiation potential of these stem cells critically rely on optimal mitochondrial function. Mitochondria-associated genes play a pivotal role in regulating stem cell activity, while Sonic Hedgehog (SHH) signaling serves as an intrinsic anti-aging mechanism [231]. SHH signaling is essential for multiple aspects of reproductive health, and its dysregulation can have detrimental effects on fertility. However, the excessive activation of SHH signaling has been shown to exacerbate endometrial fibrosis by impairing autophagy [232]. The age-related attenuation of SHH signaling diminishes stem cell activity, leading to infertility and compromised embryonic viability [231].

SERPINB2 (serpin family B member 2), also known as plasminogen activator inhibitor type 2 (PAI-2), is a serine protease inhibitor primarily recognized for suppressing urokinase-type plasminogen activator (uPA), thereby modulating fibrinolysis and extracellular matrix remodeling [231]. Beyond its protease-inhibitory role, SERPINB2 is implicated in diverse cellular processes, including survival, differentiation, adhesion, and migration. Importantly, SERPINB2 is considered a key regulator of SHH signaling, and its age-associated upregulation contributes to stem cell senescence [231]. Axin2 has been identified as a potential biomarker of endometrial stem cells, reflecting alterations in Wnt signaling that may affect stem cell function [233]. Age-related mitochondrial decline leads to stem cell exhaustion through dysregulated mitochondrial dynamics [234], persistent inflammation, oxidative stress, and impaired energy metabolism. Deficiencies in mitochondrial fusion proteins such as MFN1, MFN2, and OPA1 inhibit stem cell differentiation and regenerative capacity [234]. In addition, SASP factors secreted by aged cells disrupt the stem cell niche, thereby diminishing tissue repair capacity [137]. The excessive accumulation of ROS induces DNA damage, impairs self-renewal, and promotes differentiation and apoptosis in stem cells [235]. Progressive telomere shortening further accelerates apoptosis and cellular senescence, ultimately compromising reproductive function [236].

These age-associated changes impair the regenerative capacity of endometrial stem cells, contributing to endometrial atrophy [19], infertility, recurrent pregnancy loss, and placental dysfunction [237]. Studies in aged murine models have demonstrated that mitochondrial dysfunction leads to stem cell depletion and delays wound healing [238], suggesting that similar mechanisms may underlie stem cell decline in humans, including neural stem cell depletion [239]. Although direct evidence linking mitochondrial dysfunction to endometrial stem cell depletion in humans remains limited, emerging research indicates that the age-related decline in stem cell function plays a significant role in endometrial aging and its associated pathologies [231]. Therefore, preserving mitochondrial integrity is essential for maintaining stem cell function and mitigating premature cellular aging. Further investigations are needed to elucidate the specific molecular pathways connecting mitochondrial dysfunction, endometrial stem cell depletion, and endometrial aging.

### 4.10. Altered Intercellular Communication

During embryo implantation, precise and coordinated communication between endometrial and immune cells is essential for successful implantation, immune tolerance, and placental development, thereby establishing a receptive microenvironment [240]. Disruptions in these intercellular signaling pathways can compromise endometrial regeneration and function, ultimately impairing implantation outcomes. The underlying molecular mechanisms include hormonal regulation, cytokine-mediated signaling, gap junction interactions, and exosome-mediated communication [241]. Estrogen and progesterone regulate the expression of genes, adhesion molecules, and cytokines critical for endometrial receptivity, facilitating implantation and sustaining pregnancy [242]. Connexin 43, a pivotal component of gap junctional complexes, is predominantly expressed in stromal cells, facilitating direct intercellular communication and contributing significantly to endometrial tissue remodeling [114]. Additionally, cytokines such as TNF-α, IL-1β, IL-6, IL-8, and IFN-γ, secreted by both endometrial and immune cells, orchestrate the inflammatory response necessary for implantation [243]. These cytokines are expressed in all three major endometrial cell types: luminal epithelial cells, glandular epithelial cells, and stromal cells. Extracellular vesicles, which transport miRNAs, regulate gene expression in immune cells, modulate inflammation, and enhance tissue receptivity [244].

Age-related mitochondrial dysfunction impairs intercellular communication through several mechanisms, including the exacerbation of the senescence-associated secretory phenotype (SASP), diminished immune surveillance, disrupted interactions between stem cells and their niches, and altered signal transduction pathways driven by chronic inflammation [17]. Furthermore, mitochondrial dysfunction disturbs the metabolic balance of NAD^+^ and acetyl-CoA, interfering with epigenetic regulation and gene expression and perpetuating low-grade inflammation [245]. Oxidative stress-induced cellular damage further amplifies the SASP, activating proinflammatory signaling cascades [246]. The inflammatory cytokines and proteases secreted by senescent cells adversely affect neighboring cells and disrupt the local tissue microenvironment. As a result, intercellular communication between endometrial and immune cells is compromised, which may impair reproductive function. Thus, the dysregulated crosstalk between endometrial and immune cells may contribute to mitochondrial dysfunction, thereby accelerating reproductive aging and its associated pathologies. However, the precise role of age-related impairments in intercellular communication as a driver of reproductive decline via mitochondrial dysfunction remains an understudied area in human research.

### 4.11. Immune Regulation

The immune milieu of the endometrium is finely regulated to support embryo implantation while simultaneously protecting against infection [247]. During implantation, both innate and adaptive immune responses are modulated to create an environment conducive to sperm and blastocyst acceptance [248]. Innate immunity is primarily mediated by uterine natural killer (uNK) cells and macrophages, which also play essential roles in placental development [248]. In contrast, adaptive immunity, involving T and B lymphocytes, is tightly regulated to ensure tolerance toward the semi-allogeneic embryo [248]. The dysregulation of inflammasome activation has been implicated in reduced endometrial receptivity [249], while alterations in the Th1/Th2 cytokine balance and fluctuations in immune cell populations are commonly observed in the cases of recurrent implantation failure [250]. Moreover, immune dysregulation during pregnancy is associated with complications such as preterm birth and preeclampsia, as inflammatory imbalances negatively affect placental function and fetal development [251]. A properly balanced immune system is therefore essential for successful implantation and pregnancy, with the complex interplay between innate and adaptive immunity being critical to the prevention of infertility. Notably, the precise control of inflammatory responses is linked to improved IVF outcomes [252].

With advancing age, alterations in immune cell function contribute to a chronic proinflammatory state that adversely affects fertility. A decline in the function of endometrial immune cells, including dendritic cells, increases susceptibility to infection and impairs tissue repair mechanisms [253]. Age-related immune changes are characterized by the upregulation of chemokines such as C-X-C motif chemokine ligand 12 (CXCL12), CXCL14, and interleukin-17 receptor B (IL17RB), which have been identified in the aging endometrium [254]. The activation of IL17RB induces the expression of SASP factors, thereby accelerating cellular senescence and contributing to the decline in endometrial function [255]. These immunological alterations likely underlie the age-associated reduction in reproductive potential. In addition, mitochondrial dysfunction—through damage or apoptosis—can lead to the release of mitochondrial DNA (mtDNA) into the extracellular space [256,257,258]. Extracellular mtDNA is recognized by the NLRP3 inflammasome [259], which activates chronic inflammatory pathways [260] and disrupts intercellular communication [258]. Age-related immune dysregulation and persistent inflammation reduce endometrial receptivity and impair fertility, implantation, and embryo development [261]. Given the critical role of mitochondria in immune regulation, mitochondrial dysfunction may represent a key driver of age-related fertility decline.

## 5. Discussion

This review highlights the pivotal role of mitochondrial function in endometrial aging and its consequential impact on declining fertility. Endometrial aging is governed by a multifaceted array of factors, including abnormal endometrial thickness, dysregulated hormonal responses, chronic inflammation, cell cycle arrest, genomic instability, telomere attrition, and epigenetic modifications. The maintenance of endometrial homeostasis is critically dependent on the synergistic interplay between steroid hormone production and mitochondrial function. However, with advancing age, disruptions in hormone synthesis and mitochondrial integrity contribute to cellular senescence and diminished reproductive capacity, primarily mediated through complex interactions within the inflammatory microenvironment [102]. A moderate increase in reactive oxygen species (ROS) serves as a physiological signal that activates protective cellular mechanisms, including the upregulation of antioxidant enzymes, enhancement of DNA repair pathways, and stimulation of mitophagy to preserve mitochondrial quality. This adaptive response enhances cellular stress resistance and mitigates oxidative damage [58] (Figure 7). Within a controlled threshold, ROS may exert beneficial effects by delaying the onset of age-associated pathologies. However, progressive mitochondrial dysfunction in aging cells leads to the cytosolic release of damaged mitochondrial DNA (mtDNA), triggering chronic inflammatory cascades [262]. This decline in mitochondrial efficiency is further characterized by impaired energy metabolism, the loss of mitochondrial membrane potential, morphological abnormalities, and uncontrolled ROS overproduction. Consequently, in aged cells, excessive ROS oxidizes nucleic acids and proteins, thereby exacerbating genomic instability, accelerating telomere shortening, and inducing epigenetic alterations. These dysfunctions impair key cellular processes within the endometrium, including stem cell maintenance, and disrupt critical quality control systems such as autophagy and mitophagy. The resulting structural and functional deterioration at the cellular level culminates in widespread tissue degeneration [263]. Collectively, these degenerative changes drive cellular senescence, contribute to functional decline at the tissue level, and ultimately lead to systemic aging. Thus, endometrial aging reflects the irreversible accumulation of mitochondrial dysfunction, which substantially impairs reproductive capacity. Taken together, the decline in endometrial fertility is shaped by multiple interconnected hallmarks of aging, with mitochondrial dysfunction occupying a central role in mediating each of these processes.

Historically, endometrial dysfunction has been primarily attributed to increased ROS production and oxidative damage to mtDNA and proteins during aging. However, this view oversimplifies the complexity of mitochondrial aging. Emerging evidence suggests that mild mitohormesis may enhance endometrial resilience. Accordingly, therapeutic strategies that focus not only on mitigating oxidative stress but also on optimizing mitochondrial dynamics and quality control may offer more effective interventions against reproductive aging. In this review, we have collated recent advances for understanding endometrial cellular aging and examined the intricate relationship between mitochondrial dysfunction and reproductive senescence. Mitochondrial quality is maintained by a network of interdependent regulatory pathways that preserve organelle function and adaptability in aging cells. Further research is warranted to elucidate the precise molecular mechanisms linking mitochondrial integrity to endometrial aging and to develop targeted therapeutic strategies aimed at sustaining endometrial health across the reproductive lifespan.

## Figures and Tables

**Figure 1 ijms-26-05060-f001:**
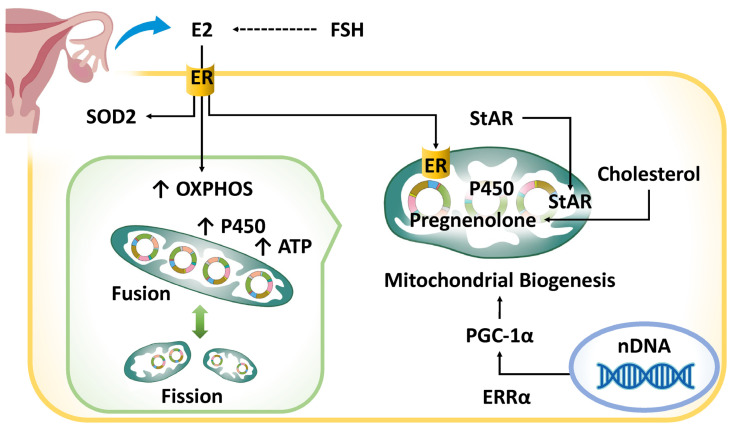
The impact of hormonal dysregulation on endometrial aging and mitochondrial dysfunction. StAR is synthesized in the cytoplasm and transported to mitochondria, where it facilitates cholesterol transfer for steroidogenesis. Mitochondria perform the rate-limiting steps of estrogen biosynthesis. ERRα, a nuclear transcription factor, regulates mitochondrial biogenesis and metabolism. Beyond estrogen production, mitochondria are key targets of estrogen, which modulates their function and cellular homeostasis. The blue arrow in the upper left corner signifies the secretion of E2 from the ovaries. ATP, adenosine 5′ triphosphate; E2, estradiol; ER, estrogen receptor; ERRα, estrogen-related receptor alpha; FSH, follicle-stimulating hormone; OXPHOS, oxidative phosphorylation; PGC-1α, peroxisome proliferator-activated receptor gamma coactivator-1 alpha; SOD, superoxide dismutase; StAR, steroidogenic acute regulatory protein.

**Figure 2 ijms-26-05060-f002:**
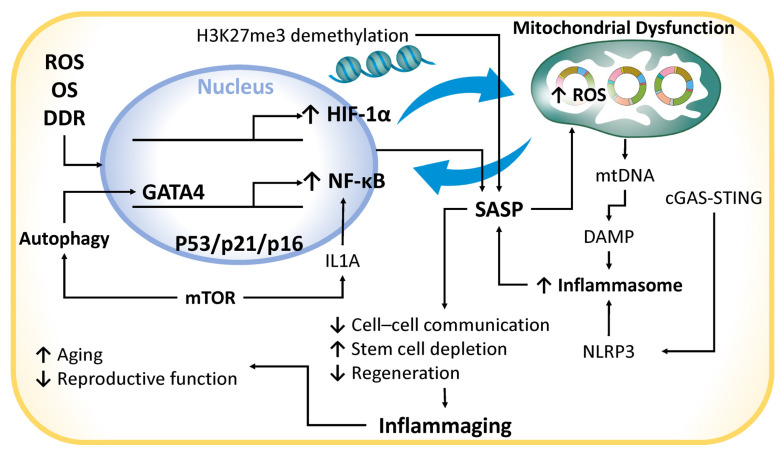
The impact of chronic inflammation on endometrial aging and mitochondrial dysfunction. SASP factor expression is regulated by pathways like HIF-1α, NF-κB, p53, p21, and p16INK4a, which influence aging and disease. The mTOR pathway drives IL1A translation, amplifying SASP production via NF-κB. The cGAS-STING pathway detects mtDNA, promoting NLRP3 inflammasome activation, a key driver of chronic inflammation. Superscript arrows denote enhancement or activation, whereas subscript arrows signify suppression or inhibition. cGAS-STING, cyclic GMP–AMP synthase–stimulator of interferon genes; DAMP, damage-associated molecular pattern; DDR, DNA damage response; GATA4, GATA-binding protein 4; HIF-1α, hypoxia-inducible factor 1-alpha; IL1A, interleukin-1α; NLRP3, NLR family pyrin domain-containing 3; mtDNA, mitochondrial DNA; mTOR, mechanistic target of rapamycin kinase; NF-κB, nuclear factor-kappa B; OS, oxidative stress; ROS, reactive oxygen species; SASP, senescence-associated secretory phenotype.

**Figure 3 ijms-26-05060-f003:**
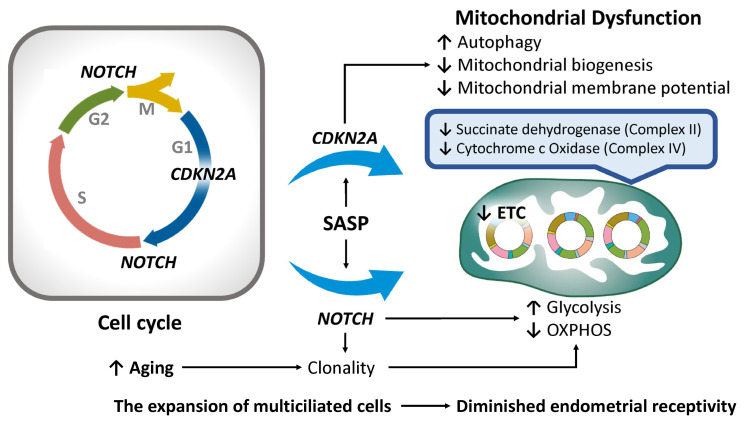
The impact of cell cycle arrest on endometrial aging and mitochondrial dysfunction. The CDKN2A gene regulates the cell cycle and mitochondrial function, while NOTCH genes modulate metabolism. Their clonal expansion may help counteract aging stress but, alongside mitochondrial dysfunction, can worsen senescence and tissue decline. Superscript arrows denote enhancement or activation, whereas subscript arrows signify suppression or inhibition. CDKN2A, cyclin-dependent kinase inhibitor 2A; ETC, electron transport chain; OXPHOS, oxidative phosphorylation; SASP, senescence-associated secretory phenotype.

**Figure 4 ijms-26-05060-f004:**
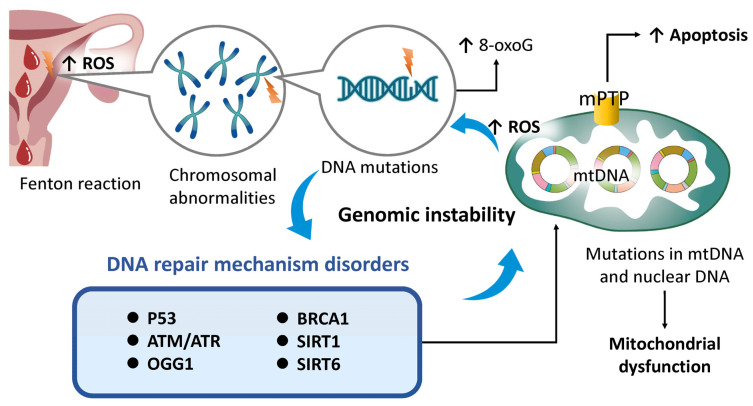
The impact of genomic instability on endometrial aging and mitochondrial dysfunction. The Fenton reaction from repeated menstruation generates ROS, causing DNA mutations that disrupt mitophagy. Over time, impaired DNA repair worsens mitochondrial dysfunction, linking genomic instability to mitochondrial decline. Superscript arrows denote enhancement or activation. 8-oxoG, 8-oxoguanine; ATM, ATM serine/threonine kinase; ATR, ATR checkpoint kinase; BRCA1, BRCA1 DNA repair associated; mPTP, mitochondrial permeability transition pore; OGG1, 8-oxoguanine DNA glycosylase 1; ROS, reactive oxygen species; SIRT1, sirtuin 1.

**Figure 5 ijms-26-05060-f005:**
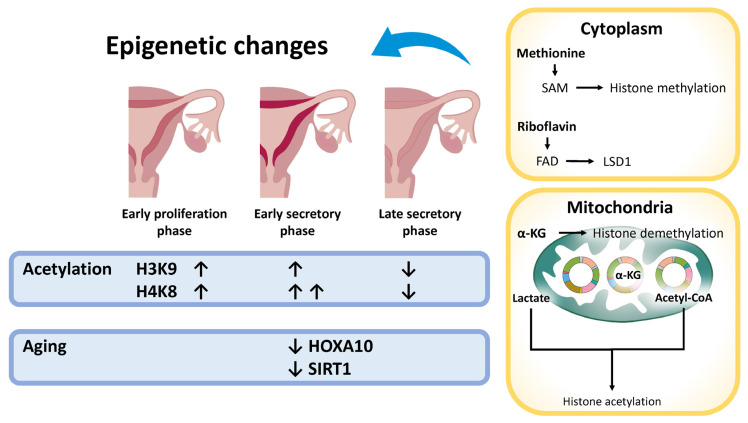
The impact of epigenetic modifications on endometrial aging and mitochondrial dysfunction. ROS impair DNA repair by methylating MSH2 and GADD45A promoters. Endometrial histone acetylation fluctuates with the menstrual cycle but declines with aging, reducing HOXA10 and SIRT1 expression and reproductive function. Metabolites like SAM, FAD, α-KG, and acetyl-CoA, obtained from diet and synthesized by mitochondria, regulate DNA methylation and histone modifications, linking metabolism to epigenetics. Superscript arrows denote enhancement or activation, whereas subscript arrows signify suppression or inhibition. A pair of superscript arrows denotes a more potent stimulus than a single arrow. Acetyl-CoA, acetyl coenzyme A; α-KG, α-ketoglutarate; FAD, flavin adenine dinucleotide; HOXA10, homeobox A10; ROS, reactive oxygen species; SAM, S-adenosylmethionine; SIRT1, sirtuin 1.

**Figure 6 ijms-26-05060-f006:**
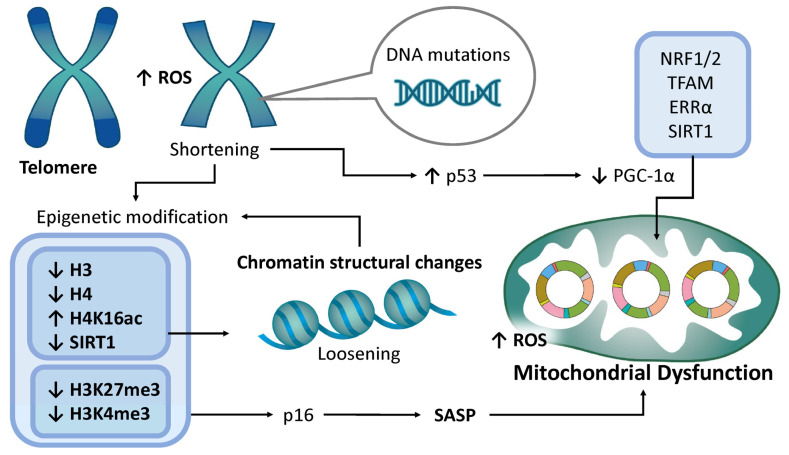
The impact of telomere attrition on endometrial aging and mitochondrial dysfunction. In senescent cells, an increase in H4K16ac and a concomitant decrease in the levels of histone deacetylase SIRT1 lead to modifications in chromatin structure. Telomere shortening exacerbates genomic instability through DNA damage and chronic inflammation, mediated by histone modifications, including methylation and acetylation. Additionally, telomere attrition induces mitochondrial dysfunction through the activation of p53 and a reduction in PGC-1α. Conversely, mitochondrial dysfunction accelerates telomere shortening by promoting oxidative stress. The deep blue regions at the terminal ends of the chromosomes represent telomeres. Superscript arrows denote enhancement or activation, whereas subscript arrows signify suppression or inhibition. ERRα, estrogen-related receptor alpha; NRE1/2, nuclear respiratory factors 1/2; ROS, reactive oxygen species; SIRT1, sirtuin 1; SASP, senescence-associated secretory phenotype; TFAM, mitochondrial transcription factor A.

**Figure 7 ijms-26-05060-f007:**
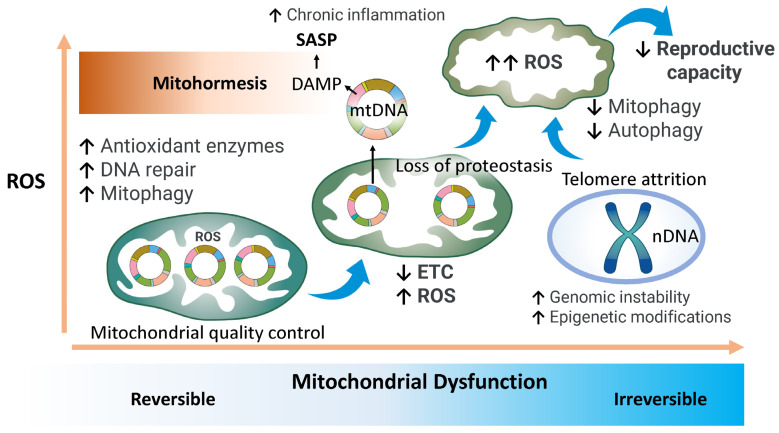
The temporal and spatial implications of disrupted mitochondrial homeostasis in endometrial subfertility. The horizontal axis of the figure represents time, while its vertical axis shows the ROS levels. Excessive ROS cause oxidative damage to DNA, proteins, and lipids, contributing to aging and cancer. However, sublethal oxidative stress can trigger adaptive responses, improving cellular resilience. Mild oxidative stress in mitochondria activates protective mechanisms, enhancing mitochondrial function and stress resistance, potentially extending lifespan. Cellular senescence halts damaged cell proliferation and acts as a tumor suppressor but leads to mitochondrial dysfunction over time. This dysfunction in the endometrium accelerates aging and age-related pathologies, with mitochondrial impairment linked to reduced reproductive capacity. Superscript arrows denote enhancement or activation, whereas subscript arrows signify suppression or inhibition. DAMP, damage-associated molecular pattern; ETC, electron transport chain; mtDNA, mitochondrial DNA; ROS, reactive oxygen species; SASP, senescence-associated secretory phenotype.

## Data Availability

No new data were created.

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
