# Peer review of "Endometrial Aging and Reproductive Decline: The Central Role of Mitochondrial Dysfunction"

_ijms, 2025, doi:10.3390/ijms26115060_

Round 1

Reviewer 1 Report

Comments and Suggestions for Authors

In present work, Kobayashi et al. try to explore endometrial aging with a focus on mitochondrial function. Hormonal dysregulation, chronic inflammation, cell cycle arrest, genomic instability, epigenetic changes, telomere attrition, and mitochondrial dysfunction contribute to structural and functional deterioration of endometria. Authors think that mitochondrial dysfunction appears central to the endometrial aging process. However, there are some questions that should be explained.

Major concerns

  1. Hormonal abnormality is a key feature of endometrial aging, and age-related endocrine alterations negatively impact endometrial physiology. Therefore, a figure is needed that hormones from different organs or tissues contribute to endometrial physiology.
  2. Figure 1, E2 and P4 are mainly form ovaries. can not secrete E2. Therefore, Figure 1 must be corrected.
  3. This manuscript only focuses on E2 and its receptors. In addition, P4 and PG are also important for endometria, but this manuscript is not included.
  4. Gut microbiota has little relation with endometria, so ‘Gut Microbiota Dysbiosis’ may be deleted.

Minor concerns

  1. Abstract section, a simple summary is needed at the end of abstract.
  2. Line 49, change ‘[4] [5],’ to ‘[4,5],’. Please check it throughout the manuscript.
  3. Scientific paper should be written in a third person manner. There are many ‘we’, which should be corrected.
  4. The reference format is not suitable for this Journal.

Comments on the Quality of English Language

The English could be improved to more clearly express the research.

Author Response

Answer to the reviewers

Manuscript ID: ijms-3598731

Title: Endometrial aging and reproductive decline: The central role of mitochondrial dysfunction

Authors: Hiroshi Kobayashi *, Miki Nishio, Mai Umetani, Hiroshi Shigetomi, Shogo Imanaka, Hiratsugu Hashimoto

Dear Editor in Chief:

IJMS

Thank you and the reviewers for the thoughtful comments and helpful suggestions on our manuscript. We have carefully considered each of the comments, made every effort to address the concerns raised, and applied corresponding revisions to the manuscript. Additionally, we have carefully revised the manuscript to ensure that the text is optimally phrased and free from typographical and grammatical errors.

The detailed, point-by-point responses to the reviewer comments are given below, whereas the corresponding revisions are highlighted to our manuscript within the document. The sentences in blue in the text were newly added.

I believe that our manuscript has been considerably improved as a result of these revisions, and hope that the revised manuscript is acceptable for publication in IJMS.

I would like to thank you once again for your consideration of our work and inviting us to submit the revised manuscript. I look forward to hearing from you.

With best regards,

Hiroshi Kobayashi, M.D., Ph.D.

E-mail: hirokoba@naramed-u.ac.jp

Point-by-point responses to reviewer comments

Reviewer 1

Comments and Suggestions for Authors

In present work, Kobayashi et al. try to explore endometrial aging with a focus on mitochondrial function. Hormonal dysregulation, chronic inflammation, cell cycle arrest, genomic instability, epigenetic changes, telomere attrition, and mitochondrial dysfunction contribute to structural and functional deterioration of endometria. Authors think that mitochondrial dysfunction appears central to the endometrial aging process. However, there are some questions that should be explained.

Comment 1:

Major concerns

Hormonal abnormality is a key feature of endometrial aging, and age-related endocrine alterations negatively impact endometrial physiology. Therefore, a figure is needed that hormones from different organs or tissues contribute to endometrial physiology.

Response 1:

Endometrial physiology is intricately regulated by a complex interplay of hormones and signaling molecules produced by various organs and tissues. Age-related endocrine alterations can disrupt this delicate balance, contributing to endometrial aging and diminished reproductive function. The hypothalamus secretes gonadotropin-releasing hormone (GnRH), which stimulates the anterior pituitary to release follicle-stimulating hormone (FSH) and luteinizing hormone (LH). These hormones regulate ovarian follicular development and ovulation, thereby influencing estrogen and progesterone production. The ovaries produce estrogen and progesterone in response to FSH and LH stimulation. Estrogen promotes the proliferation of the endometrial lining during the follicular phase, while progesterone induces secretory transformation and decidualization during the luteal phase. The endometrial tissue itself produces various cytokines, growth factors, and hormones that modulate its own function and receptivity. For instance, during decidualization, the endometrium secretes prolactin and insulin-like growth factor-binding protein 1 (IGFBP1), which are critical for embryo implantation and maintenance of early pregnancy. The adrenal glands and adipose tissue contribute to the hormonal milieu by producing androgens and estrogen precursors, which can be converted to active estrogens in peripheral tissues, including the endometrium. Thyroid hormones also play a role in modulating reproductive function, and thyroid dysfunction can adversely affect endometrial receptivity. With advancing age, alterations in the production and regulation of these hormones can lead to impaired endometrial function. For example, decreased estrogen and progesterone levels can result in inadequate endometrial proliferation and secretory transformation, respectively, compromising implantation and fertility.

I believe this statement most effectively aligns with the reviewers' expectations. However, given that the manuscript had become overly reminiscent of a textbook, I have streamlined the content and omitted explanatory components, which is why I have refrained from creating a new figure.

The following sentence has been added to "4.1. Hormonal Abnormalities":

Endometrial physiology is tightly regulated by hormones and signaling molecules originating from the hypothalamus, pituitary gland, ovaries, adrenal glands, adipose tissue, and thyroid. Gonadotropin-releasing hormone (GnRH) from the hypothalamus stimulates the release of follicle-stimulating hormone (FSH) and luteinizing hormone (LH) from the anterior pituitary, which in turn regulate ovarian estrogen and progesterone production—both essential for endometrial proliferation and decidualization [121]. Additionally, the endometrium itself secretes important implantation-related factors such as prolactin and insulin-like growth factor-binding protein 1 (IGFBP1) [121]. Aging disrupts this hormonal network, leading to reduced levels of estrogen and progesterone, impaired endometrial function, and diminished fertility.

Comment 2:

Figure 1, E2 and P4 are mainly form ovaries. can not secrete E2. Therefore, Figure 1 must be corrected.

Response 2:

Figure 1 has been modified.

Comment 3:

This manuscript only focuses on E2 and its receptors. In addition, P4 and PG are also important for endometria, but this manuscript is not included.

Response 3:

A PubMed search using the keywords “progesterone/prostaglandin,” “endometrium,” “mitochondria,” “aging/senescence” yielded no relevant publications. It has been reported that a decrease in progesterone is associated with the aging of endometrial cells, but whether this occurs via mitochondrial dysfunction remains unknown. Therefore, P4 and PG are not included in this review.

Comment 4:

Gut microbiota has little relation with endometria, so ‘Gut Microbiota Dysbiosis’ may be deleted.

Response 4:

The section "Gut Microbiota Dysbiosis" has been deleted.

Comment 5:

Minor concerns

Abstract section, a simple summary is needed at the end of abstract.

Response 5:

The following text was added: Endometrial senescence, precipitated by irreversible mitochondrial impairment, may underlie the progressive decline in reproductive potential.

Comment 6:

Line 49, change ‘[4] [5],’ to ‘[4,5],’. Please check it throughout the manuscript.

Scientific paper should be written in a third person manner. There are many ‘we’, which should be corrected.

The reference format is not suitable for this Journal.

Response 6:

We fixed them following the instructions.

I asked a native speaker friend to proofread it. I think it's much easier to read now.

Comment 7:

Comments on the Quality of English Language

The English could be improved to more clearly express the research.

Response 7:

I asked a native speaker friend to proofread it. I think it's much easier to read now.

Reviewer 2 Report

Comments and Suggestions for Authors

Review
Title; Endometrial aging and reproductive decline: The central role of 2 mitochondrial dysfunction

 Kobayashi et al. reviewed the effects of mitochondrial dysfunction on age-related declines in fertilization, implantation, and pregnancy rates. This is a useful review for those involved in reproductive medicine. This review examines a wide range of literature on gynecological diseases and focuses on mitochondrial dysfunction as the factor that most affects endometrial aging. Numerous reports have been published that mitochondrial function and aging involve various factors, such as reactive oxygen species (ROS)-related cellular damage and the breakdown of quality control through mitochondrial fission and fusion. Therefore, it is difficult to describe all of the molecular mechanisms related to mitochondrial function. As the title suggests, this review should be written with a focus on endometrial function, aging, and especially on the important molecular functions related to mitochondria and reproductive medicine. However, many of the statements in this review focuses on molecular mechanisms related to mitochondria and aging. In other words, it is insufficient to understand endometrial aging and the associated mitochondrial functions. The authors need to significantly rewrite this review paper the pathogenesis of endometrial aging and related diseases or risks such as reduced fertility from the perspective of mitochondrial dysfunction.

Other comments

Line 62; “Accumulating data highlight mitochondrial dysfunction as a key factor in reproductive aging [31-33].” The reference mentions mitochondrial dysfunction, but does not mention its relationship to reproductive aging, so this needs to be corrected.

The contents of line 137 “Autophagy” and line 154 “Mitophagy” should describe the general functional differences between Autophagy and Mitophagy. Furthermore, it would be better to separate these two from the section 2.2 Mitochondrial Quality Control and write them as 2.3.

The font size of line 33, “1, Introduction “and line 72, “2, Age-Related Changes in Mitochondrial Function” is large. In comparison, the font size of line 169, “3, Age-related regulation oh endometrial receptivity” and other titles are small font size and need to be corrected. 

The chapter "4. Properties of endometrial aging" contains too much. Furthermore, there are many explanations about the general function of mitochondria, while many explanations about molecular mechanisms that are less related to mitochondria. This chapter is redundant and difficult to understand and needs to be reconstructed into multiple chapters.

Have they already been reported for Figures 2, 3, 4, and 5 in endometrial stromal cells? If a hypothesis is included, it must be clearly stated as a hypothesis. If the hypothesis is not included, it is better to state that it is evident in mitochondria of endometrial cells.

Author Response

Answer to the reviewers

Manuscript ID: ijms-3598731

Title: Endometrial aging and reproductive decline: The central role of mitochondrial dysfunction

Authors: Hiroshi Kobayashi *, Miki Nishio, Mai Umetani, Hiroshi Shigetomi, Shogo Imanaka, Hiratsugu Hashimoto

Dear Editor in Chief:

IJMS

Thank you and the reviewers for the thoughtful comments and helpful suggestions on our manuscript. We have carefully considered each of the comments, made every effort to address the concerns raised, and applied corresponding revisions to the manuscript. Additionally, we have carefully revised the manuscript to ensure that the text is optimally phrased and free from typographical and grammatical errors.

The detailed, point-by-point responses to the reviewer comments are given below, whereas the corresponding revisions are highlighted to our manuscript within the document. The sentences in blue in the text were newly added.

I believe that our manuscript has been considerably improved as a result of these revisions, and hope that the revised manuscript is acceptable for publication in IJMS.

I would like to thank you once again for your consideration of our work and inviting us to submit the revised manuscript. I look forward to hearing from you.

With best regards,

Hiroshi Kobayashi, M.D., Ph.D.

E-mail: hirokoba@naramed-u.ac.jp

Point-by-point responses to reviewer comments

Reviewer 2

Comments and Suggestions for Authors

Review

Title; Endometrial aging and reproductive decline: The central role of 2 mitochondrial dysfunction

Comment 1:

 Kobayashi et al. reviewed the effects of mitochondrial dysfunction on age-related declines in fertilization, implantation, and pregnancy rates. This is a useful review for those involved in reproductive medicine. This review examines a wide range of literature on gynecological diseases and focuses on mitochondrial dysfunction as the factor that most affects endometrial aging. Numerous reports have been published that mitochondrial function and aging involve various factors, such as reactive oxygen species (ROS)-related cellular damage and the breakdown of quality control through mitochondrial fission and fusion. Therefore, it is difficult to describe all of the molecular mechanisms related to mitochondrial function. As the title suggests, this review should be written with a focus on endometrial function, aging, and especially on the important molecular functions related to mitochondria and reproductive medicine. However, many of the statements in this review focuses on molecular mechanisms related to mitochondria and aging. In other words, it is insufficient to understand endometrial aging and the associated mitochondrial functions. The authors need to significantly rewrite this review paper the pathogenesis of endometrial aging and related diseases or risks such as reduced fertility from the perspective of mitochondrial dysfunction.

Response 1:

Section 4 specifically addresses the molecular mechanisms driving mitochondrial dysfunction and aging, along with the associated health risks in humans, such as reduced reproductive potential. The figures have been revised accordingly. We have made every effort to exclude content not directly pertinent to the endometrium. Please see the revised parts highlighted in blue in the text. The text presents specific human examples illustrating the pathological features and risks associated with endometrial aging/senescence, including compromised fertility. If a gene or protein was identified as being associated with endometrial aging or mitochondrial dysfunction, the text specified whether the study was conducted in endometrial cells or in other cell types, such as neuronal cells. Moreover, details of human studies utilizing endometrial cells were explicitly summarized at the conclusion of each subsection.

Comment 2:

Other comments

Line 62; “Accumulating data highlight mitochondrial dysfunction as a key factor in reproductive aging [31-33].” The reference mentions mitochondrial dysfunction, but does not mention its relationship to reproductive aging, so this needs to be corrected.

Response 2:

We added corrected text to the second paragraph of the "Introduction" section.

A growing body of research suggests that mitochondrial dysfunction plays a pivotal role in aging [31–33] and constitutes a critical mechanism underlying reproductive senescence [34–36].

Comment 3:

The contents of line 137 “Autophagy” and line 154 “Mitophagy” should describe the general functional differences between Autophagy and Mitophagy. Furthermore, it would be better to separate these two from the section 2.2 Mitochondrial Quality Control and write them as 2.3.

Response 3:

We have rewritten this section per your instructions. We also divided subsection 2.2 into 2.2 and 2.3, and divided 2.3 into autophagy and mitophagy.

Comment 4:

The font size of line 33, “1, Introduction “and line 72, “2, Age-Related Changes in Mitochondrial Function” is large. In comparison, the font size of line 169, “3, Age-related regulation oh endometrial receptivity” and other titles are small font size and need to be corrected.

Response 4:

Font sizes have been standardized.

Comment 5:

The chapter "4. Properties of endometrial aging" contains too much. Furthermore, there are many explanations about the general function of mitochondria, while many explanations about molecular mechanisms that are less related to mitochondria. This chapter is redundant and difficult to understand and needs to be reconstructed into multiple chapters.

Response 5:

Chapter 4 has been divided into subsections, “4.1.” through “4.11.”, and molecular mechanisms unrelated to the endometrium have been removed whenever possible.

Comment 6:

Have they already been reported for Figures 2, 3, 4, and 5 in endometrial stromal cells? If a hypothesis is included, it must be clearly stated as a hypothesis. If the hypothesis is not included, it is better to state that it is evident in mitochondria of endometrial cells.

Response 6:

This question is related to comment 1.

Especially with regard to Figures 2-6, the text presents specific human examples illustrating the pathological features and risks associated with endometrial aging/senescence, including compromised fertility.

If a gene or protein was identified as being associated with endometrial aging or mitochondrial dysfunction, the text specified whether the study was conducted in endometrial cells or in other cell types, such as neuronal cells. Moreover, details of human studies utilizing endometrial cells were explicitly summarized at the conclusion of each subsection. The figures have also been revised accordingly. Additionally, in the subsections “Genomic instability,” “Epigenetic Modifications,” and “Telomere Attrition,”, molecular mechanisms not directly related to endometrial cells were removed.

Round 2

Reviewer 1 Report

Comments and Suggestions for Authors

Thanks for author’s responses. However, endometria include luminal epithelium, glandular epithelium and stroma. In this paper, the mitochondria are from luminal epithelium, glandular epithelium or stroma, which should be clear. Stroma cells are present in this manuscript, but epithelium cells are not. Ref. 102, human endometrial epithelium cells are involved in aging.

Comments on the Quality of English Language

The English could be improved to more clearly express the research.

Author Response

Dear reviewers,

Thank you for reviewing my paper again. We have made the necessary revisions based on your suggestions.

Best regards

Hiroshi Kobayashi

Reviewer 1

Comment 1:

Thanks for author’s responses. However, endometria include luminal epithelium, glandular epithelium and stroma. In this paper, the mitochondria are from luminal epithelium, glandular epithelium or stroma, which should be clear. Stroma cells are present in this manuscript, but epithelium cells are not. Ref. 102, human endometrial epithelium cells are involved in aging.

Response 1:

As you pointed out, in the revised version, we have described luminal epithelium, glandular epithelium, and stromal cells separately, and have highlighted the revised parts in purple. We have added the following sentence to the end of Section 3.

Senescent alterations in the endometrium influence all cellular constituents, including stromal, glandular, and luminal epithelial cells, potentially compromising the tissue’s re-generative potential and fecundity. Among these, stromal cells are pivotal for maintaining structural integrity and physiological function, and are particularly vulnerable to age-associated deterioration. In this study, the term “endometrium” refers to both stromal and epithelial components unless specified otherwise.

Comment 2:

Comments on the Quality of English Language

The English could be improved to more clearly express the research.

Response 2:

I submitted my manuscript for editing as instructed; however, only minimal revisions were made, such as the insertion of articles like "the." How can I enhance the quality of my academic English to communicate my research more effectively?

Reviewer 2 Report

Comments and Suggestions for Authors

Thank you for the revised manuscript. I have read the response letter from the authors and understand that all of the comments have been answered in the letter.

However, I could not find the additions highlighted in blue that the authors mentioned in Response 1, "Please see the revised parts highlighted in blue in the text." in the manuscript I received. 

Most of the revised manuscript is written in red and it is difficult to see where the additional text has been added. You will need to clearly indicate the added parts and resubmit the manuscript.

I think you forgot to delete "limited studies are showing" on line 733.

Author Response

Dear reviewers,

Thank you for reviewing my paper again. We have made the necessary revisions based on your suggestions.

Best regards

Hiroshi Kobayashi

Reviewer 2

Comment 1:

Thank you for the revised manuscript. I have read the response letter from the authors and understand that all of the comments have been answered in the letter.

However, I could not find the additions highlighted in blue that the authors mentioned in Response 1, "Please see the revised parts highlighted in blue in the text." in the manuscript I received.

Most of the revised manuscript is written in red and it is difficult to see where the additional text has been added. You will need to clearly indicate the added parts and resubmit the manuscript.

Response 1:

If you open a Word manuscript with "no change history or comments," the revisions will be displayed. The PDF manuscript has been sent to the secretariat.

Comment 2:

I think you forgot to delete "limited studies are showing" on line 733.

Response 2:

I forgot to delete it.

Round 3

Reviewer 2 Report

Comments and Suggestions for Authors

Thank you for your revised manuscript.

I think this is a very useful review article when considering the relationship between endometrial aging and mitochondria.